# Microbiota-Associated HAF-EVs Regulate Monocytes by Triggering or Inhibiting Inflammasome Activation

**DOI:** 10.3390/ijms24032527

**Published:** 2023-01-28

**Authors:** Emilia Nunzi, Letizia Mezzasoma, Ilaria Bellezza, Teresa Zelante, Pierluigi Orvietani, Giuliana Coata, Irene Giardina, Krizia Sagini, Giorgia Manni, Alessandro Di Michele, Marco Gargaro, Vincenzo N. Talesa, Gian Carlo Di Renzo, Francesca Fallarino, Rita Romani

**Affiliations:** 1Department of Medicine and Surgery, University of Perugia, Polo Unico Sant’Andrea delle Fratte, P.e Lucio Severi 1, 06132 Perugia, Italy; 2Department of Obstetrics and Gynecology, University Hospital of Perugia, Sant’Andrea delle Fratte, P.e Lucio Severi 1, 06132 Perugia, Italy; 3Department of Molecular Cell Biology, Institute for Cancer Research, Oslo University Hospital, The Norwegian Radium Hospital, 0379 Oslo, Norway; 4Department of Physics and Geology, University of Perugia, Via Pascoli, 06123 Perugia, Italy; 5Department of Obstetrics, Gynecology and Perinatology IM Sechenov First State University, 117997 Moscow, Russia

**Keywords:** extracellular vesicles, amniotic fluid, LPS, metagenomic analysis

## Abstract

In pregnancy, human amniotic fluid extracellular vesicles (HAF-EVs) exert anti-inflammatory effects on T cells and on monocytes, supporting their immunoregulatory roles. The specific mechanisms are still not completely defined. The aim of this study was to investigate the ability of HAF-EVs, isolated from pregnant women who underwent amniocentesis and purified by gradient ultracentrifugation, to affect inflammasome activation in the human monocytes. Proteomic studies revealed that HAF-EV samples expressed several immunoregulatory molecules as well as small amounts of endotoxin. Surprisingly, metagenomic analysis shows the presence of specific bacterial strain variants associated with HAF-EVs as potential sources of the endotoxin. Remarkably, we showed that a single treatment of THP-1 cells with HAF-EVs triggered inflammasome activation, whereas the same treatment followed by LPS and ATP sensitization prevented inflammasome activation, a pathway resembling monocyte refractories. A bioinformatics analysis of microbiota-HAF-EVs functional pathways confirmed the presence of enzymes for endotoxin biosynthesis as well as others associated with immunoregulatory functions. Overall, these data suggest that HAF-EVs could serve as a source of the isolation of a specific microbiota during early pregnancy. Moreover, HAF-EVs could act as a novel system to balance immune training and tolerance by modulating the inflammasome in monocytes or other cells.

## 1. Introduction

Amniotic fluid (AF) surrounds the embryo and fetus during development and performs multiple functions. It protects the fetus physically in case the maternal abdomen is subjected to trauma. Moreover, AF contains several proteins, electrolytes, immunoglobulins, and vitamins from the mother that are useful to the fetus. The components of AF can be divided into three distinct groups: (i) An insoluble fraction, containing cellular elements; (ii) a soluble fraction consisting of carbohydrates, proteins, lipids, electrolytes and metabolites; and (iii) a fraction that containing extracellular vesicles (EVs) [1]. In 2005, Underwood et al. reported the presence of antimicrobial, immunomodulatory, and growth-promoting activities in AF [2,3,4]. Accordingly, the immunomodulatory properties of AF have been demonstrated in studies of preterm pigs with necrotizing enterocolitis, in which AF suppressed pro-inflammatory responses [5]. Extracellular vesicles (EVs) are a highly heterogeneous population of lipid bilayer-vesicular structures that are released by almost all living cells and can perform a wide range of critical biological functions [6]. As carriers of biologically active molecules such as proteins, lipids, and nucleic acids, EVs mediate cell-to-cell communication and perform various functions by delivering their cargo to specific cell types under normal and pathological conditions, including angiogenesis, cellular differentiation, osteogenesis, inflammation and cancer [7,8,9,10,11].

In pregnancy, EVs are essential for diverse physiological processes [6] and may play a role in the successful development of the fetus [12]. Perinatal cells of various tissues and their extracellular vesicles have been shown to play an immunoregulatory role in both the innate and adaptive immune systems [13]. In particular, placental amniotic cells or their secretome have been shown to suppress T-cell proliferation, reduce the release of inflammatory cytokines, and prevent the differentiation of T cells into Th1 and Th7 [13,14,15,16,17]. In addition, EVs derived from placental trophoblasts are continuously released into maternal plasma and appear to play a critical role in regulating maternal immune response and adaptation to pregnancy [18]. Previously, we had also shown that EVs released from amniotic fluid stem cells have anti-inflammatory effects on T cells [19] and more recently also on monocytes [20]. Accordingly, studies by Del Rivero et al. [21] demonstrated that AF-EVs have a suppressive effect on T-cell proliferation, T-cell activation, and on the release of pro-inflammatory cytokines. Recently, AF-EVs have been used as a novel therapy in the treatment of acute respiratory distress syndrome to reduce the effects of COVID-19-induced cytokine storm, supporting their immunoregulatory effects [22,23].

In most women, spontaneous labor occurs at the time of delivery in the absence of intra-amniotic infection and is therefore considered a sterile inflammatory process. The mechanisms responsible for the sterile inflammatory process of labor at term are not fully understood. There is in vivo evidence of inflammasome activation during spontaneous labor at term potentially explaining the sterile inflammatory processes in term and preterm labor [24]. Inflammasome activation is now recognized as key mediator of acute and chronic inflammatory responses. Inflammasome activation can be triggered by a wide variety of molecules, including ligands for molecular patterns associated with endogenous hazards/damage (DAMPs) and molecular patterns associated with exogenous pathogens (PAMPs), such as lipopolysaccharide (LPS). They tightly regulate the production of proinflammatory cytokines such as IL-1β and IL-18 by activating caspase-1 through interaction with ASC (apoptosis-associated speck-like protein containing a carboxy-terminal CARD), an adaptor protein that bridges NOD-like receptors (NLRs) and caspase-1 [25]. The canonical NOD-like receptor 3 (NLRP3) protein is the most studied and best known inflammasome receptor that recognizes multiple microbial and endogenous danger signals [26]. IL-1β maturation by caspase-1-mediated cleavage represents a key inflammatory event. Acute LPS stimulation, followed by ATP, is frequently used to activate the NLRP3 inflammasome in macrophages [27]. Interestingly, it has been observed that the ability of LPS to license NLRP3 is transient, as prolonged LPS exposure triggers regulatory mechanisms to dampen NLRP3 activation [28]. Endotoxin (lipopolysaccharide [LPS]) tolerance is characterized by reduced responsiveness of specific types of cells especially monocytes and macrophages to subsequent challenge with LPS. In in vitro animal and human experiments, tolerance to LPS is associated with attenuated proinflammatory cytokines, such as interleukin-1 and interleukin-6 and increased production of anti-inflammatory cytokines, including interleukin-10 [29,30]. There are several lines of evidence that endotoxin tolerance leads to selective suppression of a number of tolerizeable genes after restimulation with the same or similar stimulants [31,32]. Indeed, unregulated innate immunity can lead to maladaptive immune responses that promote chronic inflammatory states.

Recently, the “sterile womb” paradigm has been challenged because there is experimental evidence that the first contact of the offspring with the microbiota occurs before birth in uncomplicated full-term pregnancies. Establishing a healthy microbiota early in life could be a key element in reducing the burden of disease later in life. It is now known that the environment of a healthy AF is colonized by microbes, which is considered a prerequisite for fetal immune maturation, metabolic and hormonal homeostasis [33]. Under physiological conditions, the placental microbiome harbors non-pathogenic commensals, including Firmicutes, Tenericutes, Proteobacteria, Prevotella, Neisseria, Bacteroidetes, and Fusobacteria, but also potential pathogenic species such as *Escherichia coli* [33]. Since its detection is based on molecular techniques, there is an ongoing scientific debate as to whether the placental microbiome contains a viable microbiota or only microbial components [33]. Recently, by analyzing human fetal and placental tissue in the second trimester of pregnancy, Mishra A et al. identified live bacterial strains, capable of inducing activation of T cells in the fetal mesenteric lymph node [34]. However, whether a microbiome can be detected in EVs, which are important for cell-to-cell communication, and may be enriched in early pregnancy AF, is currently unknown.

Thus, the main objective of this study was to analyze whether human amniotic fluid-derived EVs (HAF-EVs) could negatively affect the activation of the inflammasome complex in a human monocytic THP-1 cell line. The demonstration of the ability of HAF-EVs to both train immunity and induce tolerance in THP-1 cells prompted us to investigate the presence of a microbiota in the HAF-EVs.

## 2. Results and Discussion

### 2.1. Characterization of HAF-EVs

Stem cells derived from embryonic annexes (e.g., placenta, Wharton’s umbilical cord) release (EVs) with immunomodulatory properties that overlap with those of the cells from which they are derived [13]. We hypothesize that pools of HAF-EVs most likely mediate immune regulatory effects. Since we had previously shown that human amniotic stem cell-derived extracellular vesicles (HASC-EVs) exert immunoregulatory functions [19] and attenuate inflammasome activation in human THP-1 monocytes [20], here we investigated whether HAF -EVs could also affect the inflammasome machinery.

EVs were isolated by differential centrifugation from human amniotic fluid of twenty-eight pregnant women aged 35–43 years who volunteered to participate in the experimental project (MAF2019). (Figure 1a). We obtained two fractions, designated HAF-P10-EVs (sedimented at 10,000× *g*) and HAF-P100-EVs (sedimented at 100,000× *g*). The presence of membrane vesicles was confirmed by Scanning electron microscopy (SEM) imaging of surfaces and particle size analysis (Figure 1b,c). In addition, nanoparticle tracking analysis (NTA) allowed us to evaluate the concentration, average size and size profile for the two subsets of EVs (Figure 1d). The mean diameter was 227 ± 12 nm for HAF-P10 and 187 ± 14 nm for HAF-P100, while the concentration was 1.06 × 10^11^ ± 0.054 particles/µg proteins and 4.38 × 10^11^ ± 0.53 particles/µg proteins for HAF-P10 and HAF-P100, respectively. Both HAF–EVs expressed the typical EV marker proteins, including Alix (ALG-2 interacting protein X), CD81 (Tetraspanin-28), ARF6 (ADP-ribosylation factor 6) and TSG101 (Tumor susceptibility gene 101) (Figure 1e).

To better characterize and evaluate the functions of HAF–EVs, we performed a qualitative proteomic analysis. The complete list of proteins found to be co-expressed by HAF–EVs from four different donors is shown in Appendix A. Of particular interest was the presence of proteins with bacteriostatic properties, such as: Transferrin, Bone marrow proteoglycan, Serpin 1 [35], and BPI fold-containing family A member 1 [36,37,38].

Further analysis of the proteins expressed by the two subsets of EVs revealed that most of these proteins were described in the Vesiclepedia database (Figure 2a), and more specific analysis confirmed that they were predominantly found in extracellular vesicles (Figure 2b). Further analysis of potentially involved biochemical pathways revealed several biological functions, including some related to immune activation and regulation (Figure 2c). Overall, these data show that AF contains different types of EVs. Based on proteomic studies, these results suggest that EVs from AF may function as regulators of immune responses.

### 2.2. Bimodal Regulation of the Inflammasome by HAF-EVs in THP-1 Cells

To investigate the potential immunomodulatory role of HAF–EVs, THP-1 cells were pretreated with HAF–P10- or HAF–P100- (100 µg/mL) for 1 h, and then stimulated with LPS and ATP (LPS + ATP) [20,39]. Surprisingly, we found that pretreatment with both HAF–EV types alone triggered inflammasome platform activation, as indicated by significantly increased expression of mature caspase-1 (Figure 3a) and IL-1β (Appendix A). In contrast, the same pre-treatment resulted in inhibition of caspase-1 and IL-1β activation when followed by LPS + ATP treatment (Figure 3a and Appendix A). The same effect of HAF-EVs on inflammasome activation or suppression could also be observed in purified CD14+ human monocytes (Appendix A). The observed opposite effect of HAF-EVs on the inflammasome platform led us to hypothesize that pretreatment with HAF-EVs might immunologically train THP-1 cells, leading to a state of tolerance to subsequent inflammatory stimuli. 

Since it is known that a small amount of LPS can induce a tolerogenic effect characterized by a cellular refractoriness to a second LPS stimulus [40,41], we analyzed the possible presence of endotoxin in HAF-EVs. We found a level of endotoxin equal to 2 ± 0.13 pg/µL, 3 ± 0.09 pg/µL in HAF-P10 and HAF–P100 (HAF-EVs), respectively. Notably, the PBS samples, used as controls, and subjected to the same environment and procedures as the amniotic fluid, including differential centrifugation steps, had undetectable endotoxin levels.

Given the presence of LPS in both subfractions of EVs, we focused only on P100 EVs in the following experiments. Next, we tested whether the small amount of LPS found in the HAF-EVs could mimic the effect of inflammasome activation found by treatment with HAF -EVs. To this end, THP-1 cells were treated with 100 µL volume (e.g., 10 µg protein, 6.55 × 10^13^ ± 0.23 number of particles) of HAF–EVs corresponding to 300 pg LPS or with an equal amount of LPS alone and assayed for inflammasome activation. Remarkably, we observed that 300 pg LPS mirrored the effect of HAF–EVs on caspase-1 activation both in the absence and presence of LPS + ATP treatment (Figure 3b).

Recently, a growing body of studies has been focused on delineating the complex mechanisms underlying the regulation of inflammasome signaling [42]. Our data show that HAF–EVs can represent an additional mean to regulate inflammatory immune responses in human monocytic cells via inflammasome modulation. Negative modulation of the inflammasome by EVs from different sources has been claimed in numerous cellular systems and attributed to various molecular mechanisms [43,44,45,46,47,48]. Accordingly, we have previously shown that EVs, isolated from human amniotic stem cells (HASCs–EVs), contain immunoregulatory cargo molecules capable of altering the phenotype of T lymphocytes [19] and that HASC–EVs, act as independent metabolic units and inhibit inflammasome activation in human THP-1 monocytes [20]. Surprisingly, the results of this study suggest that EVs in AF may play a role in both inhibition and activation of an innate immune response because they may contain different biological molecules.

In this study, we found a bimodal action of HAF–EVs in inflammasome regulation human monocytes. A single treatment with HAF–EVs triggered inflammasome activation in THP-1 cells, whereas the same treatment followed by LPS and ATP exposure prevented inflammasome activation. Moreover, we demonstrated, for the first time, the presence of significantly low levels of endo-toxin in HAF–EVs, but not in control EVs. Remarkably, pretreatment of THP-1 cells with this low concentration of LPS, as found in HAF–EVs, mimicked the effect of HAF–EVs in activating and suppressing inflammasome activation, which is similar to an effect associated with monocyte refractoriness or endotoxin tolerance. These data have some implications and suggest that low levels of endotoxin may be present in AF, which may be important for immune training and the regulation of innate immune responses in the growing fetus.

Endotoxin tolerance is characterized by reduced reactivity to subsequent LPS exposure. At present, our data do not want to show that HAF–EVs regulate inflammasome activation in THP-1 through associated LPS alone. Rather, regulation may be more complex because EVs isolated from various biological fluids contain cargo characterized by high levels of different biomolecules [49]. Additional molecules contained in EVs might therefore contribute to further fine tune the inflammatory or tolerance state in monocytic cells. Indeed, the “lipopolysaccharide-tolerant” phenotype is characterized by inhibition of several signaling pathways, including lipopolysaccharide-stimulated tumor necrosis factor production, altered release of interleukin-1 and interleukin-6, enhanced activation of cyclooxygenase-2, inhibition of mitogen-activated protein kinase activation, and altered translocation of nuclear factor κB [50]. An inappropriate activation of caspase-1/IL-1β pathway has been associated with the pathogenesis of a wide range of auto-inflammatory, auto-immune and metabolic disorders and has been very recently involved in recurrent miscarriage [51].

Since the fine regulation of these mechanisms is critical for homeostatic immune responses, preventing excessive inflammatory responses, and promoting the resolution of inflammation [52], it is possible that the ability of HAF–EVs to stimulate THP-1 cells is a key mechanism for training the inflammasome in the first place and promoting its inhibition during a subsequent challenge with inflammatory stimuli. Indeed, this is the first finding linking EVs isolated from amniotic fluid to the regulation of the inflammasome through cooperation with associated bacterial products. Therefore, these findings may be of greater importance as these mechanisms may help the fetus tolerate the nascent microbiota [31], which is also involved in the development of the immune system.

The identification of LPS in HAF–EVs prompted us to investigate the presence of microbiome niches by metagenomics studies in HAF–EVs.

### 2.3. Microbiota Species Variants in HAF-EVs

For this purpose, we analyzed the HAF–EVs of 28 HF samples from 16–17 weeks pregnant women. At the same time, we analyzed 12 sterile injectable water samples that served as blanks. The blanks and HAFs were ultra centrifuged, and microbial DNA was extracted from the EV pellets and analyzed by using 16S rRNA gene sequencing approach. The rarefaction curve shows the species richness of each sequenced sample com-pared to the corresponding sequencing depth. All samples with good sequencing quality were sufficiently sequenced, as the species richness shows a flat line. To compare the diversity indices between the two groups, samples were rarefied to 12,000 reads (Appendix A).

To assess whether the composition of the microbiome of the HAF–EVs and the blank samples showed significant differences in richness or abundance at any taxonomic level, the observed features and Shannon alpha indices were evaluated for each group at each taxonomic level (e.g., from strain to species) and for species variances (SV).

We demonstrated a low but consistent microbial signal across different samples and the average values of both indexes grow with the taxonomic level and show a maximum at SV (Figure 4a and Appendix A).

In addition, the alpha index of the AF for each taxonomic level had a higher mean alpha index value than that of the blank samples, suggesting that the amniotic samples had, on average, a greater number of taxa and a less homogeneous compositional distribution than the control samples. In particular, we found that the differences in the number of taxa between groups from phylum to the family and at SV were significant (Figure 4a,b). However, the clustering of the observed traits at the genus and species levels was not significant. This indicates that the amniotic fluid samples have a significant number of features that are not included in the blanks, as also shown by the heatmap at SV (Figure 4c and Appendix A). We also found 30 prevalent features in HAF–EVs compared to the blanks (Figure 4d).

The composition of the microbiota in HAF–EVs samples at the phylum level included seven major phyla: Bacteroidetes, Firmicutes, Proteobacteria, Actinobacteria, Cyanobacteria, Deinococcota, and Fusobacteria (Appendix A). 

Fusobacteria and Deinococcota characterize the Amnio samples. Deinococcus species (es-pecially D. radiodurans) have been isolated from particularly extreme environments, such as radioactive waste sites and hot springs, but also in the human gut microbiome and especially in the niche of the gastric microbiome [51]. More importantly, Fusobacteria, Gram-negative bacilli, especially Fusobacterium nucleatum, is an oral bacterium commonly associated with preterm birth. Oral *fusobacteria* can colonize the placenta, but the mechanisms leading to local localization in the placenta remain unclear. Recently, binding between lectin and Fusobacterium was demonstrated [52]. Here, we show that Fusobacterium is present in EVs isolated from the AF of healthy women, although we did not investigate the presence of viable bacteria in this study.

The existence of an amniotic microbiota, i.e., a viable microbial cell community in mammals, is quite controversial [53]. Recently, Corynebacterium tuberculostearicum and Lactobacillus murinus were shown to be more abundant in the HAF compared to controls [54]. Importantly, viable cultures could only be reproduced for *L. murinus*. We also performed additional studies to investigate possible metabolic pathways in the microbiota of HAV -EVs. We analyzed the EC/KO database for HAF SVs to virtually identify the metabolic properties of the HAF microbiome. Interestingly, we show, in Figure 5, that the most abundant metabolic features are related to the biosynthesis of LPS (Appendix A) and amino acids. 

Overall, our data suggest that it is very possible that microbial components are present and exposed to the fetus in utero, but HAF may not be a source of fully viable bacteria. Accordingly, various studies have recently suggested that certain bacterial molecules may cross the placental barrier and make their way to fetal organs [55,56,57,58,59]. Remarkably, the fetal microbiome was recently profiled in several fetal organs using 16S rRNA gene sequencing, and a low but stable microbial signal was detected in fetal organs such as the gut in the second trimester of pregnancy. Moreover, in the same study viable and cultivable bacterial strains such as *Staphylococcus* and *Lactobacillus* were detected in fetal tissues, which induced antigen-specific memory T cells in mesenteric lymph nodes. In agreement with this study, our results support the role of HAF and EVs in particular as a possible new pathway for fetal tissue microbial exposure during pregnancy [34]. HAV–EVs containing specific microbiota could contribute to the fine-tuning of the fetal immune system. This hypothesis is consistent with recent reports suggesting that priming of the fetal immune system begins during pregnancy [58,59,60,61,62]. Moreover, our study demonstrates for the first time that such microbial presence, particularly in the form of secretory means (i.e., EVs), is involved in both priming and suppression of at least the inflammasome pathway in monocytes, placing EV-associated microbial factors in the context of fetal innate immune priming and tolerance, the latter being a concept not previously explored in fetal immunity. This study provides novel evidence for the presence of microbial molecules in EVs from HAF in early gestation, however there are several limitations to be addressed in future studies. First, the precise source of bacteria in AF remains unknown. Second, sequencing the maternal microbiome, possibly at the same time, would be an important issue to address.

Moreover, further studies are required to explore the microbial interactions in human fetal organs and the synergy with cytokines or other factors in shaping both adaptive and innate immune responses.

## 3. Materials and Methods

### 3.1. EVs Derived from HAF Isolation 

Human amniotic fluids (HAF) were isolated from 16–17-week pregnant women (aged 35–43 years) who underwent amniocentesis during routine prenatal diagnosis. The study was approved by Region Ethics Committee (CER) Umbria code 3599/19(MAF2019), and each participant provided informed consent for the secondary use of amniotic fluid samples.

EV isolation was performed with a sequential centrifuge protocol according to [63]; briefly, human amniotic fluids (HAF) were pooled, centrifuged at 300× *g* for 10 min, followed by centrifugation at 2000× *g* for 20 min. At third centrifugation at 10,000× *g* for 45 min was used to collect the P10 pellet. The supernatant was subjected to an ultracentrifugation step at 100,000× *g* for 60 min in Optima TLX ultracentrifuge with 60Ti rotor (Beckman Coulter) to obtain the p100 pellet. P10 and P100 pellets were washed with PBS added with antibiotics (1% penicillin and streptomycin) and centrifuged again at 100,000× *g* for 60 min. The pellets were suspended with PBS added with antibiotics. We used sterile certified centrifuge tubes free of any detectable DNA, DNase, RNase, PCR inhibitors and endotoxins (Beckman Coulter, Brea, CA, USA) and sterile practices.

### 3.2. Nanoparticle Tracking Analysis (NTA)

HAF–P10 and HAF–P100 pellets were resuspended in PBS (filtered through a 0.02 µm Anotop 25 filter) to obtain a concentration within the recommended range (2 × 10^8^–1 × 10^9^ particles per mL). Samples were vortexed for 1 min and then loaded into a NS500 instrument (Malvern Instruments Ltd., Worcestershire, UK). For each sample, 5 videos of 60 sec were acquired and processed using the NTA2.3 software. Particles moving under Brownian motion were tracked and their hydrodynamic diameter was calculated using the Stokes-Einstein equation.

### 3.3. Scanning Electron Mycroscopy (SEM) Analysis 

HAF-EVs were fixed in 1.5% glutaraldehyde for 15 min, at room temperature, washed with water, sedimented on glass coverslips and then allowed to dry at room temperature. SEM images were obtained using a field emission gun electron scanning microscope (LEO 1525 Zeiss; Thomwood, NY, USA) with Cr metallization using a high-resolution sputter 150T ES-Quorum apparaus (24 s, sputter at a current of 240 mA). The chromium thickness was ~10 nm.

### 3.4. Protein Content Analysis

#### 3.4.1. SDS-PAGE and in-Gel Digestion

Proteomic analysis P100 and P10 fractions was performed according to Mezzasoma et al., 2022 [20]. Briefly, P10 and P100 proteins were separated on 6–15% polyacrylamide gels and stained with Oriole Fluorescent gel stain (Bio-Rad, Hercules, CA, USA). 

The whole gel was sliced in 20 bands for each lane. The bands were washed and dehydrated twice in 100 mM ammonium bicarbonate (ABC)/50% acetonitrile (ACN), in 50 mM (ABC)/50% (ACN) and in 25 mM (ABC)/50% (ACN). Cysteine bonds were reduced with 10 mM DTT/25 mM ABC at 56 °C for 1 h and alkylated with 50 mM iodoacetamide/25 mM ABC at room temperature for 45 min in the dark. The bands were washed in water, dried in a vacuum centrifuge and incubated overnight with 6.25 ng/mL trypsin at 37 °C and covered with ABC to allow protein digestion. Peptides were extracted twice in 0.1% formic acid/60% ACN, dried in vacuum centrifuge and resuspended in 100 µL of loading buffer (5% ACN/0.1% formic acid).

#### 3.4.2. LC-MS/MS Analysis and Database Searching

Oligopeptides were separated by using a ProteomeX apparatus (Thermo Scientific, San Jose, CA, USA) equipped with a 100 μL loop and a Hypersil-Keystone BioBasic C18 capillary column (0.18 × 100 mm) and configured in the Protein ID mode. The oligopeptide mixtures were separated with a gradient of ACN containing 0.1% formic acid (solvent B), at a flow rate of 2 μL/min and eluted with a 35 min linear gradient from 5 to 60% solvent B, followed by a 5 min linear gradient from 60 to 80% solvent B and 8 min isocratic elution with 80% solvent B. 

Eluted oligopeptides were electrosprayed into the LCQ Deca-XPPlus ion-trap mass spectrometer. Database searching was performed by the MASCOT software version 2.2, accessed on 6 June 2017 (http://www.matrixscience.com/, accessed on 1 January 2020) [64].

#### 3.4.3. Western Blot Analysis 

Total THP-1 cell lysates (10 µg) were separated by 12% sodium dodecyl sulfate-polyacrylamide gel electrophoresis (SDS-PAGE) and transferred on nitrocellulose membrane. Non-specific binding sites were blocked in Roti-Block (Roth GmbH Karlsruhe, Germany) for 1 h at room temperature. The membranes were blotted overnight at 4 °C with the following anti-human Abs diluted in Roti-Block: anti-Caspase-1 polyclonal antibody (pAb) (#2225); anti-IL-1β (3A6) pAb (#12242). After washing with TBST, blots were incubated for 1 h at room temperature with the appropriated HRP-conjugated secondary Abs and revealed using the enhanced chemi-luminescence (ECL) system (Amersham Pharmacia Biotech, Uppsala, Sweden). Membranes were stripped and re-probed with anti-β-actin mAb (I-19) antibody (Santa Cruz Biotechnology, CA) as loading control. Densitometric analyses were performed with ImageJ software.

#### 3.4.4. THP-1 Cell Culture, Human Monocyte Purification and Treatments

Human THP-1 monocytes were purchased from American Type Culture Collection (ATCC, USA) and routinely maintained at 37 °C in 5% CO_2_ in RPMI 1640 supplemented with 10% heat inactivated FBS, L-glutamine, 1 mM sodium pyruvate, non-essential amino acids, 1% of penicillin/streptomycin. THP-1 cells (3 × 10^6^ cells/well) were plated in 12-well culture dishes, pretreated for 1 h with 100 μg/mL HAF–derived EV-P10 (HAF–P10) or EV-P100 (HAF–P100). THP-1 cells were subsequently primed with 10 µg/mL LPS for 20 min and then activated with 5 mM ATP for 40 min. The concentration of EVs has been chosen on the basis on previous experiences [20]. At the end of the treatments, total cell lysates were prepared using RIPA buffer with protease and phosphatase inhibitor cocktail.

Human monocytes were isolated from buffy coats of healthy donors by Ficoll-Paque™ (Cytiva Life Sciences™) density gradient centrifugation. Total PMBCs were depleted of CD3+, CD16+, CD19+, CD56+, CD123+, CD235a+ cells by using biotin antibodies. Human PBMCs were incubated with anti-biotin antibodies for negative selection is following by incubation with streptavidin. After incubation, labeled cells were purified through a magnetic separation column for negative selection (LD column Miltenyi Biotec). The purity of human CD14+ CD16- cells were analyzed by flow cytometry and cells with a purity higher than 90% were used for stimulation.

### 3.5. Endotoxin Assay

EVs samples were tested for their endotoxin level with Pierce™ LAL Chromogenic Endotoxin Quantitation Kit (Thermo Scientific, San Jose, CA, USA). 

It is a sample and sensitive assay for the detection of endotoxin lipopolysaccharide (LPS), the membrane component of gram-negative bacteria. The quantitative assay uses amebocyte lysates derived from blood of the horseshoe crab to quantitate endotoxin in protein, peptides, antibodies or nucleic acid samples. HAF-EVs were resuspended in 100 μL and then diluted 1:100 for the assay. Amoebocyte lysate was added to each well according to the instructions and incubated for time 1 h. At the end of this period, the chromogenic substrate was added and incubated for 6 min, the reaction was stopped by adding 25% acetic acid. The absorbance at 405 nm is then read.

### 3.6. Microbiome Analysis in Human Amniotic Fluid 

We analyzed 28 human amniotic fluid (HAF), from 16–17 weeks pregnant women (aged 35–43 years). We simultaneously analyzed 12 sterile injectable preparation water samples used as environmental blanks. The blanks and HAF samples were ultra-centrifuged at the same time to samples, at 100,000× *g*. DNA was extracted from the pellet of approximately 15 mL of HAF ultra centrifuged at 100,000× *g* for 60 min and washed with 10 mL of sterile PBS, using the Qiagen kit.

Sequencing was outsourced as a service to the LGC group (GmbH, Hilden, Germany), which proceeded as follows:

Amplicon type: Archaeal 16S: nested PCR − 341F-1061R (20 cycles) + 515FY-926R (20 cycles)

341F-1061R

5′ CCTACGGGNGGCWGCAG 3′

5′ CRRCACGAGCTGACGAC 3′

515FY-926R

5′ NNNNNNNNNNGTGYCAGCMGCCGCGGTAA 3′

5′ NNNNNNNNNNCCGYCAATTYMTTTRAGTTT 3′

Amplicon sequencing (NGS2407 & NGS2636)

The PCRs included about 1–10 ng of DNA extract (total volume 5 µL), 15 pmol of each forward primer and reverse primer in 20 µL volume of 1 × MyTaq buffer containing 1.5 units MyTaq DNA polymerase (Bioline GmbH, Luckenwalde, Germany) and 2 µL of BioStabII PCR Enhancer (Sigma-Aldrich, Milan, Italy). For the preamplification with 341F-1061R, the PCRs were carried out for 20 cycles, using the following parameters: 1 min 96 °C pre-denaturation; 96 °C for 15 s, 50 °C for 30 s, 70 °C for 90 s. 5 µL of the pre-amplified PCR product was used for the final amplification with the barcoded primer 515FY-926R. For each sample, the forward and reverse primers had the same 10-nt barcode sequence. PCRs were carried out for 20 cycles using the following parameters: 1 min 96 °C pre-denaturation; 96 °C denaturation for 15 s, 55 °C annealing for 30 s, 70 °C extension for 90 s, hold at 8 °C. DNA concentration of amplicons of interest was assessed by gel electrophoresis. About 20 ng amplicon DNA of each sample were pooled for up to 48 samples carrying different barcodes.

The amplicon pools were purified with one volume Agencourt AMPure XP beads (Beckman Coulter, Inc., Brea, CA, USA) to remove primer dimer and other small mispriming products, followed by an additional purification on MiniElute columns (QIAGEN GmbH, Hilden, Germany). 

About 100 ng of each purified amplicon pool DNA was used to construct Illumina libraries using the Ovation Rapid DR Multiplex System 1–96 (NuGEN Technologies, Inc., Redwood City, CA, USA). Illumina libraries (Illumina, Inc., San Diego, CA, USA) were pooled and size selected by preparative gel electrophoresis. Sequencing was done on an Illumina MiSeq using V3 Chemistry.

All sequences were deposited in NCBI’s Sequence Read Archive (SRA) submission number SUB12577267, BioProject accession PRJNA928589 Microbiota Amniotic Fluid 2019 (MAF2019).

### 3.7. Bioinformatics and Statistical Analyses

All libraries for each sequencing lane were demultiplexed using the Illumina bcl2fastq v2.20 software (Illumina 2019). Barcodes were clipped from the sequence and reads with final length <100 bases were discarded and then primer sequences were detected, clipped and turned into forward–reverse primer orientation after removing the primer sequence using cutadapt. All sequenced files were subjected to a quality control procedure using FASTQC software [65] and then were imported in the next-generation microbiome bioinformatics platform Qiime2 platform (version 2022.2) [66] in a genomic cloud-computing environment based on [54], and oriented for biological nano-communication systems in blood vessels for early tumor medical diagnosis [67].

At first, paired-end sequences were denoised, dereplicated, filtered by both any phiX reads and chimera (consensus) using q2-dada2 quality control method [68] for detecting and correcting (where possible) Illumina amplicon sequence data. Sequence error profiles is employed to obtain putative error-free sequences, referred to as either sequence variants (SVs) or 100% operational taxonomic units (OTUs). Forward and reverse sequences were truncated at position 250 and 150, respectively, due to decrease in quality and also to the first instance of a quality score less than or equal to two. Forward and reverse reads with errors higher than five and three, respectively, were discarded and only reads with a minimum overlap of 12 nt were retained and joined.

SVs were assigned taxonomy using a machine learning algorithm based on the Naive Bayes classifier model trained on the Silva138 99% database trimmed to the V4–V5 region of the 16S. The supervised trained classifier was then applied to the obtained SVs for mapping them to taxonomy.

A phylogenetic tree was constructed via sequence alignment with MAFFT [69], by filtering the alignment and by applying FastTree [70] to generate the tree.

The analysis of the rarefaction curves of the Shannon index indicated a good sequencing quality as the richness index does not increase significantly with the sampling depth for each sample. In order to ease the comparison of microbial composition between the two groups of amniotic liquid and blanks, samples were normalized by rarefying sequencing to 12,000 reads. Moreover, in order to evaluate the composition of the microbial community of low abundant taxa accurately, a taxon was regarded as detected in a sample if it was counted at least 10 times in that sample, thus setting the resolution of composition analysis equal to 0.08%. Taxa with an abundance per sample lower than the given resolution were counted in the global community but not considered for differential analyses in order to avoid non-reliable comparisons of rare taxa between groups that could lead to biased analyses [71].

Further analyses were conducted using both Qiime2 (ver. 2022.2) platform and R version 4.2.1 (23 June 2022) in RStudio 2022.02.3+492 “Prairie Trillium” Release (1db809b8323ba0a87c148d16eb84efe39a8e7785, 20 May 2022) for Ubuntu Bionic Mozilla/5.0 (Windows NT 10.0; Win64; ×64; rv:108.0) Gecko/20100101 Firefox/108.0.

The within sample alpha-diversity was determined using Observed_feature and Shannon diversity indexes estimated by using the QIIME2 platform. Corresponding statistical significances in sample groups comparison were determined using a Kruskal-Wallis test [70]. In order to assess significant differences in microbial profiles (beta-diversity), statistical analysis based on the distance matrix was used by setting non-parametric permutational multivariate analysis of variance within QIIME2 (permanova, permdisp and adonis) with a *p* value of <0.05 through 9999 permutations. Jaccard and Bray-Curtis and unifrac (weighted and unweighted) distances between samples (Faith, Minchin, and Belbin 1987) were used. Principal coordinates analysis (PCoA) was applied to distance matrices in order to separate quantitatively all sources contributing to the beta diversity. PCoA was limited to the first three components, thus allowing visualization of the most effective relationships contributing to diversity between groups of samples. Permutational MANOVA [61] with 999 permutations was used to test significant differences between sample groups based on both Jaccard and Bray-Curtis distance matrices. 16S SVs were agglomerated into Phylum, Class, Order, Family, Genus and Species levels within QIIME2 for evaluating the corresponding taxonomic abundance.

The Phylogenetic Investigation of Communities by Reconstruction of Unobserved States (PICRUST2) pipeline was applied to predict metagenome functions from 16S metagenomic samples. In particular, the enzyme-catalyzed reaction (EC number), functional gene content based on KEGG database annotations for reference genomes (KEGG Orthology), and metabolic pathway abundances of microbial communities using the pathway rules from MetCyc database were predicted with PICRUSt2 [71]. Sequenced samples were provided as ASV abundance tables (rarefied at 12,000 reads) and files with representative sequences. To estimate the extent of microbial metabolic pathway representation across amniotic liquid samples PICRUSt2 was applied in order to identify a number of metabolic pathways present in the samples.

Grubbs test in the R package was used for evaluating outliers.

## 4. Conclusions

The results of this study demonstrate, for the first time, that EVs isolated from amniotic fluid from the third trimester of pregnancy contain DNA from specific bacterial variants, implying the presence of a specific microbiota in AF at the early stage of fetus development. Remarkably, due to the presence of endotoxin from this microbiota, AF-EVs exert dual activity by both activating and suppressing the inflammasome in human monocytes, resembling an endotoxin tolerance effect. These findings need to be considered when studying microbiota species during pregnancy and their impact on fine-tuning the immune responses of the growing fetus. Indeed, changes in AF microenvironmental factors or maternal immune factors may affect the quality of this microbiota and impact both differentiation and immune cell function, potentially affecting tolerance and immunity after birth.

## Figures and Tables

**Figure 1 ijms-24-02527-f001:**
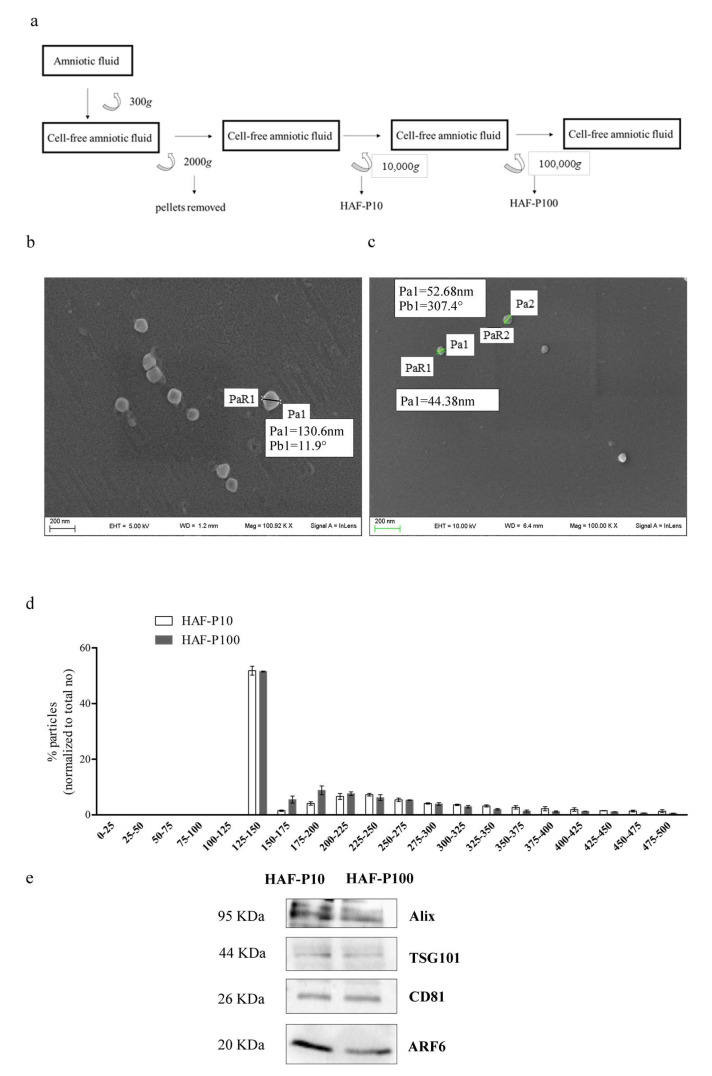
Characterization of HAF–EVs. (**a**) Flowchart of isolation procedures by differential centrifugation. (**b**,**c**) SEM images of HAF–EVs, (**d**) Profile of size distribution by nanoparticle tracking analysis. (**e**) Western blotting for EV markers. We evaluated both the presence of specific markers of EVs (Alix 95 kDa, TSG101 45 kDa and CD 81 22/26 kDa;) The HAF–EVs (P10, P100) for Western blotting analysis were from the same fresh preparation and the same amount of protein was loaded. The images and data shown are examples of a single experiment, but they are representative of at least three independent experiments.

**Figure 2 ijms-24-02527-f002:**
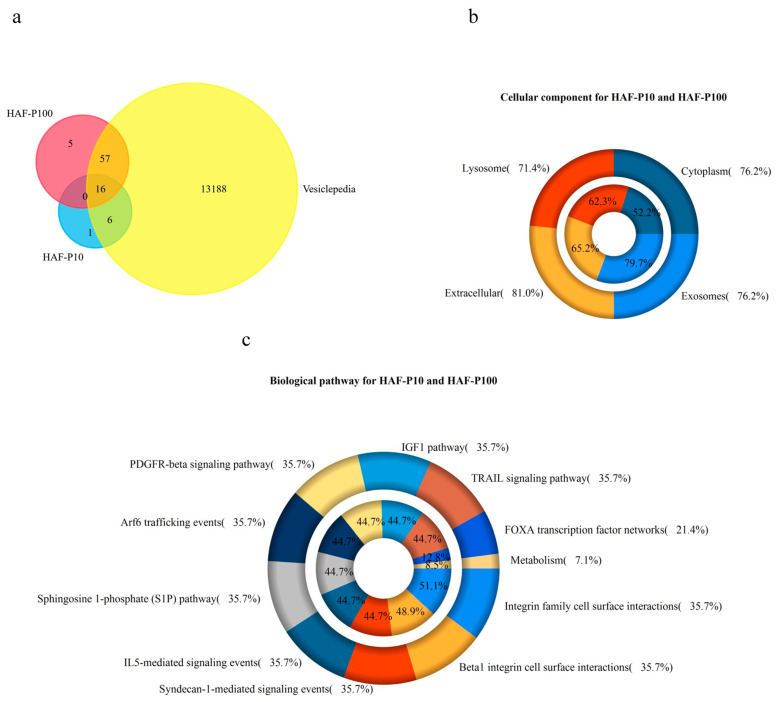
Proteomics Analysis. Bioinformatic analysis of the proteomics data was performed by means of FUNRICH software. (**a**) Venn diagram showing comparative analysis between the proteins identified in both HAF–EV subsets and the Vesiclepedia database. (**b**) The doughnut chart shows in which major cellular components the proteins identified in HAF vesicles were found. HAF–P10 on outer chart and HAF–P100 on inner chart. (**c**) The chart shows the principal pathway for HAF–EVs. HAF–P10 on the outer chart and HAF–P100 on the inner chart. The results are representative of three independent experiments.

**Figure 3 ijms-24-02527-f003:**
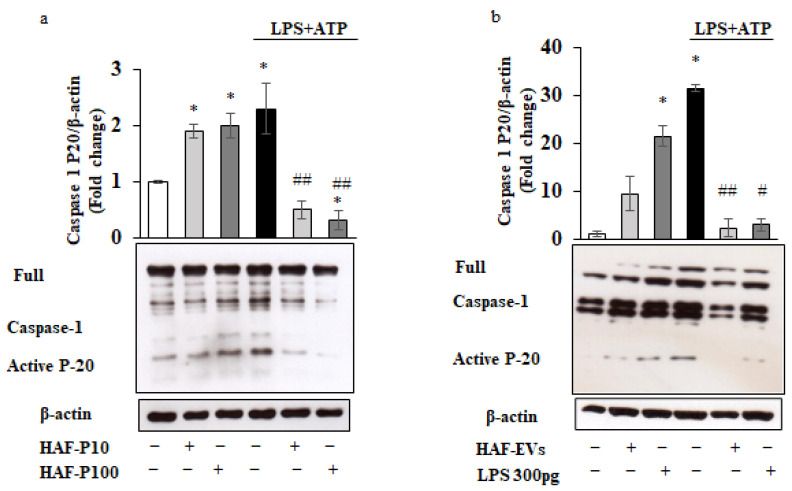
Regulation of the Inflammasome by HAF–EVs in THP-1 cells. THP-1 cells were pre-treated with 100 µg/mL of HAF–EVs (P10 or P100) for 1 h (**a**), and with 100 µg/mL of HAF–EVs (P10 and P100 pooled) or 300 pg LPS for 1 h (**b**) and subsequently primed with 10µg/mL LPS for 20 min and then activated with 5 mM ATP for 40 min (LPS + ATP). Cell lysates were immunoblotted for Caspase-1 (A, B). β-actin was used as a loading control. Representative Western blots images are shown. Histograms represent densitometric quantification and indicate the mean ± SD of at least n = 2 independent experiments. * *p* < 0.05 versus untreated cells., * *p* < 0.01, versus untreated cells. # *p* < 0.05, ## *p* < 0.001 versus LPS + ATP treated cells.

**Figure 4 ijms-24-02527-f004:**
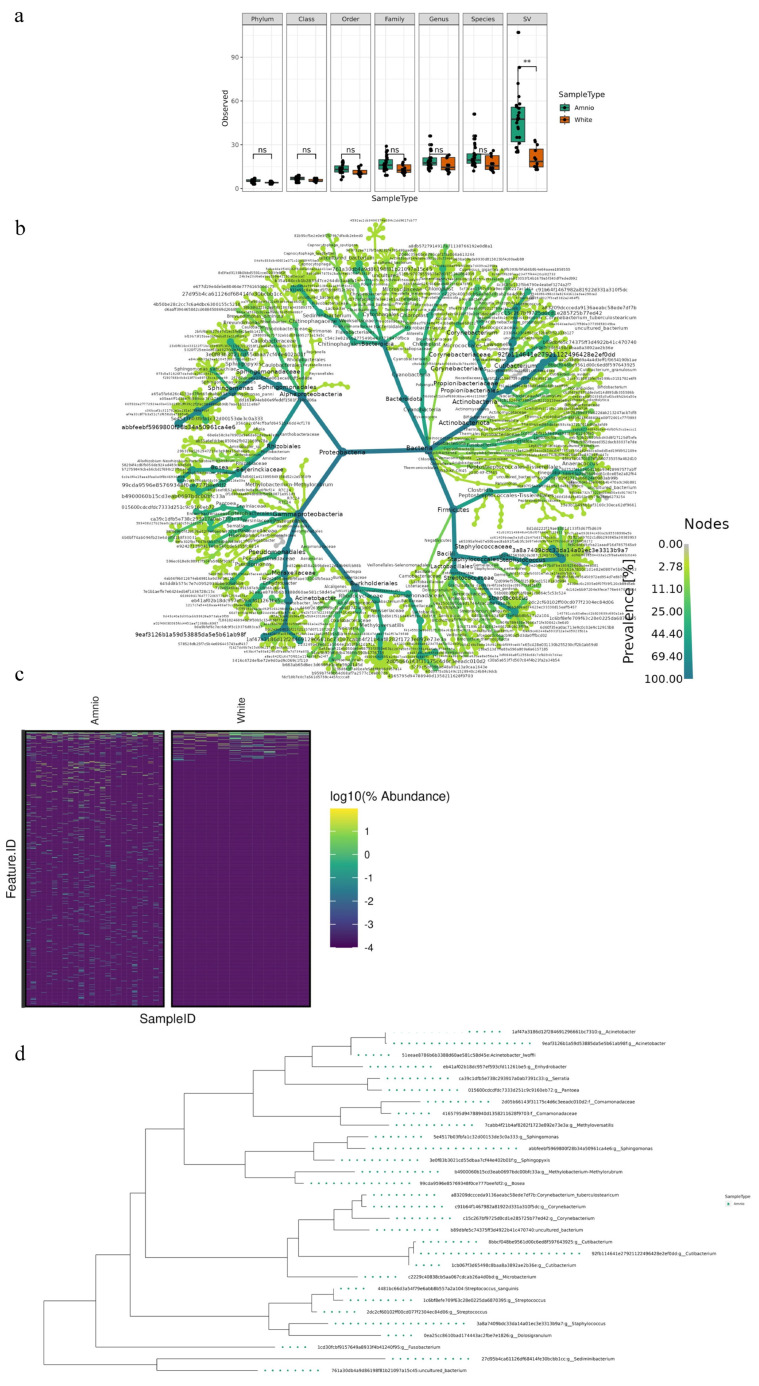
Microbiota composition in HAF–EVs. (**a**) Observed alpha index evaluated in each group (green: HAF; red: control samples) at each taxonomic level as indicated by the corresponding label. Comparison between groups (Kruskall Walli test) is also reported. ns not significant, ** *p* < 0.01. (**b**) Heat tree of sequence variant (SV)level. (**c**) Heatmap of samples grouped by sample type. The relative abundance is expressed as the logarithm of the percentage value and the taxa have been sorted by mean abundance in the blanks. A pseudo-count of 0.00001% was used on percentage values. The dark tile highlights the absence of the corresponding feature. The illuminated tile indicates the presence of the corresponding feature to an abundance value based on the colors of the legend. (**d**) Phylogenetic tree of the 30 most prevalent features in the amniotic samples.

**Figure 5 ijms-24-02527-f005:**
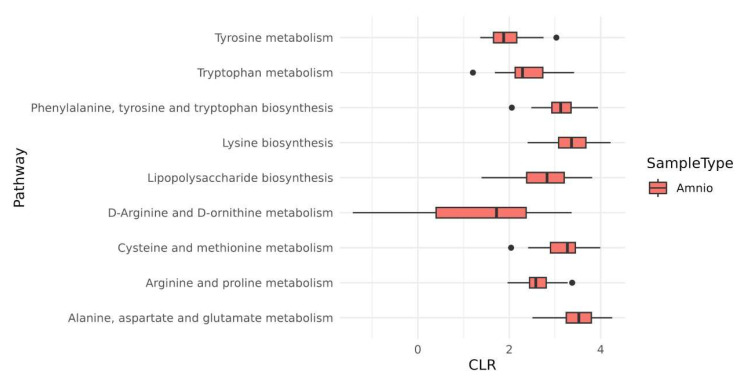
Metabolic functions of HAF microbiota. Some of the pathways identified by functional genome analysis are represented.

## Data Availability

Data supporting reported results can be found: NCBI’s Sequence Read Archive (SRA) submission number SUB12577267 BioProject accession PRJNA928589 Microbiota Amniotic Fluid 2019 (MAF2019).

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
