# Peer review of "Microbiota-Associated HAF-EVs Regulate Monocytes by Triggering or Inhibiting Inflammasome Activation"

_ijms, 2023, doi:10.3390/ijms24032527_

Round 1

Reviewer 1 Report

Comments on ijms-2168666

In this study, the author has studied “Microbiota-associated HAF-EVs regulate monocyte by triggering or inhibiting Inflammasome activation.” This is an engaging article with a robust methodology that purposefully questions our knowledge of the subject. However, the presentation of results is somewhat confusing, and the readability of the discussion could be improved. Addressing both these issues will make this interesting paper more impactful. The English language used in the manuscript needs major improvements as some punctuation and grammatical mistakes are present. Experimental designs required more clarity. Moreover, research models are not discussed in an understandable manner, which reflects that the author needs a more comprehensive way of thinking. It is obvious that the quality of the manuscript does not fulfill the standards of the journal. Therefore, it should be reconsidered after major revision.

Specific comments:

1.      Page 1, line 32-33: “A bioinformatics analysis of microbiota-HAF-EVs functional pathways confirmed the presence of enzymes for endotoxin biosynthesis...” Please add the names of enzymes.

2.      Please add more strong keywords and avoid the words used in the title.

3.      Page 1, 53-54: “all living cells and can perform a wide range of critical biological functions [4]As 53 carriers of biologically active molecules such as proteins...” Please add a full stop after ‘critical biological functions [4]’.

4.      Page 2, line 68: “Accordingly, studies by Del Rivero et al [18] demonstrated…” Somewhere authors use italics ‘et al’ and somewhere like ‘et al.’ Please use some trends throughout the manuscript.

5.      Page 2, line 76-77: “There is an in vivo evidence of inflammasome activation during spontaneous…” Please italicize ‘in vivo’ and follow this trend throughout the manuscript.

6.      Page 3, line 110: “but also potential pathogenic species such as Escherichia coli.” The scientific names should be in italics and follow this trend throughout the manuscript.

7.      Page 3: What is the research gap and novelty of the present study?

8.      Page 3, line 131-132: “fluid from pregnant women aged 35-43 years who volunteered to participate in the experimental project (MAF2019).” Please add the sample size.

9.      Page 3, line 81: “pretreated with HAF-P10- or HAF-P100- (100 μg/ml) for 1h …” Please convert ‘100 μg/ml’ to ‘100 μg/mL’ and follow this trend throughout the manuscript. Also, there is a space between a value and a unit (1 h).

10.  Most of the figures are not readable, and please increase the resolution of the figure’s minimum to 300 DPI.

11.  The results and discussion section needs revision. The discussion needs professional English editing, and please revise it carefully to make it standard. Please focus on the main topic during the discussion. An excellent discussion contains an accurate statement of the results, the relevance, and importance of the results, suitable comparisons to similar studies, alternative explanations of the findings, known limitations, and suggestions for future research.

12.  Please provide more detail in section 2.5 or merge it with another section.

13.  Please add a section of conclusions with some strong recommendations.

14.  Authors are advised to proofread the manuscript to overcome grammatical mistakes.

15.  Authors are advised to revise headings and subheadings.

16.  Many references are outdated; please revise them and add updated data.

Author Response

In this study, the author has studied “Microbiota-associated HAF-EVs regulate monocyte by triggering or inhibiting Inflammasome activation.” This is an engaging article with a robust methodology that purposefully questions our knowledge of the subject. However, the presentation of results is somewhat confusing, and the readability of the discussion could be improved. Addressing both these issues will make this interesting paper more impactful. The English language used in the manuscript needs major improvements as some punctuation and grammatical mistakes are present. Experimental designs required more clarity. Moreover, research models are not discussed in an understandable manner, which reflects that the author needs a more comprehensive way of thinking. It is obvious that the quality of the manuscript does not fulfill the standards of the journal. Therefore, it should be reconsidered after major revision. 

We thank the reviewer for revising our paper and propose insightful suggestions.

Specific comments:

  1. 1.Page 1, line 32-33: “A bioinformatics analysis of microbiota-HAF-EVs functional pathways confirmed the presence of enzymes for endotoxin biosynthesis...” Please add the names of enzymes.

Answer 1: A new Figure (i.e., Supplemental Figure 6) has been included that lists all the genes for the endotoxin biosynthesis that have been found by bioinformatic analysis of microbiota-HAF-EVs functional pathways.

  1. Please add more strong keywords and avoid the words used in the title.

Answer 2: At the suggestion of the reviewer, two new additional keywords have been included

  1. Page 1, 53-54: “all living cells and can perform a wide range of critical biological functions [4]. As 53 carriers of biologically active molecules such as proteins...” Please add a full stop after ‘critical biological functions [4]’.

Answer 3: a full stop has been added.

  1. Page 2, line 68: “Accordingly, studies by Del Rivero et al [18] demonstrated…” Somewhere authors use italics ‘et al’ and somewhere like ‘et al.’ Please use some trends throughout the manuscript.

Answer 4: We have revised the text et al. and used the same trends throughout the manuscript.

  1. Page 2, line 76-77: “There is an in vivo evidence of inflammasome activation during spontaneous…” Please italicize ‘in vivo’ and follow this trend throughout the manuscript.

Answer 5: We have revised the text in vivo and used the same trends throughout the manuscript.

  1. Page 3, line 110: “but also potential pathogenic species such as Escherichia coli.” The scientific names should be in italics and follow this trend throughout the manuscript. 

Answer 6: As suggested by the reviewer we updated the scientific names of bacteria species in italic.

  1. Page 3: What is the research gap and novelty of the present study?

Answer 7: In the revised form of the paper we have included a statement highlighting the gap and novelty of the present study.

  1. Page 3, line 131-132: “fluid from pregnant women aged 35-43 years who volunteered to participate in the experimental project (MAF2019).” Please add the sample size.

Answer 8: We included in the text the sample size.

  1. Page 3, line 81: “pretreated with HAF-P10- or HAF-P100- (100 μg/ml) for 1h …” Please convert ‘100 μg/ml’ to ‘100 μg/mL’ and follow this trend throughout the manuscript. Also, there is a space between a value and a unit (1 h).

Answer 9: We have revised the text “mL” and used the same trend throughout the manuscript.

  1. Most of the figures are not readable, and please increase the resolution of the figure’s minimum to 300 DPI.

Answer 10: We have increased the resolution of the images to at least 300DPI.

  1. The results and discussion section needs revision. The discussion needs professional English editing, and please revise it carefully to make it standard. Please focus on the main topic during the discussion. An excellent discussion containsan accurate statement of the results, the relevance, and importance of the results, suitable comparisons to similar studies, alternative explanations of the findings, known limitations, and suggestions for future research.

Answer 11:  We apologize for not describing our research findings clearly enough. In the current version of the paper, we have reorganized the results and discussed the main findings in light of known data from the field. We have also better highlighted the potential implications and limitations of the study. In addition, we have made suggestions for future studies.

  1. Please provide more detail in section 2.5 or merge it with another section

Answer 12: We thank the reviewer for this comment. We first noted that the section “2.5” should be written as “3.5”, since it belongs to section 3. In the revised version, we have also included more details about the method for measuring the amount of endotoxin in EVs

  1. Please add a section of conclusions with some strong recommendations.

Answer 13: We included a conclusion section highlighting major findings of our study and implications.

  1. Authors are advised to proofread the manuscript to overcome grammatical mistakes.

Answer 14: We apologize for some typos in the paper, we have made effort to correct them.

  1. Authors are advised to revise headings and subheadings.

Answer 15: As suggested by the reviewer some eading and subheadings were revised.

  1. Many references are outdated; please revise them and add updated data.

Answer 16: We have included more updated references in the introductory section.

Reviewer 2 Report

The manuscript is an important contribution to the field. Authors should work on the following suggestion: 

- Increase the font size in Fig1 for better visibility. 

- There is no information about patient consent and project id for the same.

- Fig 2S: Calling blank samples as blank will be more appropriate then calling them white. 

- To have more physiological relevance, authors should try to include data showing the effect of HAF EVs on inflammasome response in human blood monocytes (CD14+ cells). 

Author Response

We thank the reviewer for appreciating our study and revising our paper.

  1. Increase the font size in Fig1 for better visibility.

Answer 1:  We increased the font of Fig. 1 for better visibility.

  1. There is no information about patient consent and project id for the same.

Answer 2:  Information about the paper consent and project ID is included in the method section and it is highlighted for reference.

  1. Fig 2S: Calling blank samples as blank will be more appropriate then calling them white.

Answer 3: As suggested we renamed the white as blank.

  1. To have more physiological relevance, authors should try to include data showing the effect of HAF EVs on inflammasome response in human blood monocytes (CD14+ cells).

Answer 4:  in the revised form of the paper, new data on the effects of EVs from AF on monocyte inflammasome activation are now presented in Figure S2. Remarkably, EVs from AF show a similar bimodal effect in activating and suppressing the inflammasome in human CD14+ monocytes as in the THP-1 cells.

Round 2

Reviewer 1 Report

The authors have carefully addressed all the comments. So, the manuscript should be accepted after minor revision. 

1. The conclusion should be in a single paragraph.

Reviewer 2 Report

Authors have adequately responded to my queries. I recommend for the acceptance of the manuscript in the current form.